# Toxicity Assessment of *Curculigo orchioides* Leaf Extract Using *Drosophila melanogaster*: A Preliminary Study

**DOI:** 10.3390/ijerph192215218

**Published:** 2022-11-18

**Authors:** Sharanya Kushalan, Leonard Clinton D’Souza, Khyahrii Aloysius, Anurag Sharma, Smitha Hegde

**Affiliations:** 1Nitte (Deemed to Be University), Nitte University Centre for Science Education and Research (NUCSER), Division of Bioresource and Biotechnology, Kotekar-Beeri Road, Deralakatte, Mangaluru 575018, India; 2Nitte (Deemed to Be University), Nitte University Centre for Science Education and Research (NUCSER), Division of Environmental Health and Toxicology, Kotekar-Beeri Road, Deralakatte, Mangaluru 575018, India

**Keywords:** *Curculigo orchioides*, leaf extract, development, wing defect, reproductive toxicity, *Drosophila*

## Abstract

*Curculigo orchioides* is used in Indian and Chinese traditional medicinal systems for various health benefits. However, its toxicological effects are mostly unknown. This study assesses the potential toxicity of aqueous leaf (A.L.) extract of *C. orchioides* using *Drosophila melanogaster* as an experimental model. Preliminary phytochemical tests were followed by the Fourier transform infrared (FTIR) tests to identify the functional group in the A.L. extract of *C. orchioides*. *Drosophila* larvae/adults were exposed to varying concentrations of *C. orchioides* A.L. extract through diet, and developmental, lifespan, reproduction, and locomotory behaviour assays were carried out to assess the *C. orchioides* toxicity at organismal levels. The cellular toxicity of A.L. extract was examined by analysing the expression of heat shock protein (*hsps*), reactive oxygen species (ROS) levels, and cell death. The FTIR analysis showed the presence of functional groups indicating the presence of secondary metabolites like saponins, phenolics, and alkaloids. Exposure to A.L. extract during development resulted in reduced emergence and wing malformations in the emerged fly. Furthermore, a significant reduction in reproductive performance and the organism’s lifespan was observed when adult flies were exposed to A.L. extract. This study indicates the adverse effect of *C. orchioides* A.L. extract on *Drosophila* and raises concerns about the practice of indiscriminate therapeutic use of plant extracts.

## 1. Introduction

The use of plant-based products to prevent or treat various diseases or maintain health has surged worldwide. A report on the USA population underlines the intake of botanical supplements among all age groups of men and women [1]. A survey conducted in the Arab and U.K. states that around 80% of the population uses plants as self-medication to gain multiple health benefits [2,3]. The report suggests that worldwide herbal medicine consumption was valued at approximately USD 230.03 billion in 2021 and predicts a rise in the market size to USD 430.05 billion in 2028 [4]. Factors such as low-cost alternatives to allopathic medicine, efficacy, easy availability, and the belief in fewer side effects are the most common reason for using herbs as medicine [5]. A misconception among the general population is that plant-based medicines are generally safe as they have an ancient history of treating various illnesses. Plants have a wide range of phytochemicals such as phenolics, flavonoids, alkaloids, saponins, and terpenoids; and synergistically or additively may also elicit harmful health effects. Along this line, several reports have raised the safety concerns associated with their short and long-term uses in the recent past. For instance, studies have demonstrated the cytotoxic, genotoxic, mutagenic, reproductive, behavioural, and developmental toxicity of plants/extracts in various experimental model systems [6,7,8,9,10]. Therefore, it is pertinent to determine medicinal plants’ toxic properties before advocating their usage.

*Curculigo orchioides* Gaertn, a member of hypoxidaceae, is a well-known medicinal plant. It is commonly called “Kali musli” in India and “Xian mao” in China. It is one of the common herbs consumed in many south Asian countries like India, China, and Nepal [11]. The plant is now listed as endangered [12]. The plant’s rhizome is used in Ayurvedic and traditional Chinese medicines. In Ayurveda, rhizomes are utilized in Rasayana (anti-aging), Vrushya (aphrodisiac), and Brimhana (weight loss), whereas in China, it is used in the treatment of menstrual irregularities, menstrual cramps, and amenorrhea, in addition to strengthening the spleen, kidney, and bones [13]. The main phytoconstituents of *C. orchioides* are glycosides, alkaloids, saponins, and polysaccharides, which were isolated from the rhizome or the whole plant [14]. Among the isolated compounds, Curculigoside, a phenolic glycoside isolated from the plant’s rhizomes, is the major bioactive component and has a variety of biological activities such as neuroprotective and antiosteoporotic [15,16]. Moreover, the medicinal value of different extracts of *C. orchioides* has also been confirmed as anti-inflammatory, anti-diabetic, aphrodisiac, anti-oxidant, and anti-microbial [17,18,19,20,21]. Regardless of the health benefits of *C. orchioides*, its therapeutic implementation may be hindered due to inadequate information on its toxicity. As a result, the purpose of this study was to assess the toxicity of *C. orchioides* aqueous leaf extract at the organismal level (development, reproduction, behaviour, and lifespan) and the cellular level (stress genes expression, oxidative stress, and apoptosis) using the *Drosophila melanogaster* model.

*Drosophila* is the closest invertebrate model to humans, with conserved evolutionary genetics and developmental biology. High reproductive capacity, faster development, shorter life span, and conserved biological processes make it an ideal model for toxicological assessment [22]. Moreover, *Drosophila* has been recommended for experimental studies by the Organization for Economic Cooperation and Development (OECD) and the European Centre for the Validation of Alternative Methods [23]. To the best of our knowledge, this is the first report on the toxicity of Aqueous leaf (A.L.) extract *C. orchioides* at the organismal level. Our research indicates that the supplementation of A.L. extracts to Drosophila larvae for 96 h does not elicit gut toxicity, however, the supplementation of A.L. extract to larvae or adult flies for the long-term, significantly hinders the development, reproduction, and survival of the organism. This study would help guide the recommendations for supplementation of A.L. extracts in the future.

## 2. Materials and Methods

### 2.1. Materials

FTIR (Bruker Optick Gm BH, Ettlingen, Germany), Leica stereomicroscope (Model: S9D), RNAiso Plus reagent (#9108, TAKARA, Shiga, Japan), MicroDrop spectrophotometer (Multiscan Sky, Thermo Scientific, Waltham, MA, USA), qPCR (QuantStudioTM 5 Real-Time PCR system, Applied Biosystems, Waltham, MA, USA), Olympus BX53 fluorescent microscope, T.B. Green master mix (RR820A, TAKARA, Japan), DCFH-DA; Sigma, St. Louis, MO, USA, 2201608), anti-caspase 3 (1:500, #9661S, R&D systems, Minneapolis, MN, USA) and secondary antibody (#A11030; Goat anti-mouse Alexa flourTM 546; 1:500 dilution).

### 2.2. Collection of Plant Material and Extraction

The leaves of mature flowering *C. orchioides* were collected during the months of August to October from the Kasaragod District of Kerala (12.5885° N, 75.2628° E). Following the protocol of Aloysius et al. [24], the extraction was carried out. Briefly, fresh leaves were cleaned in sterile distilled water blended with a known volume of water (1:1), and shaken for 24 h at 120 rpm. The aqueous extract was filtered at the end of 24 h using a fine muslin cloth. The clear filtrate was dried using a flash evaporator at 42 °C and the pressure maintained at 25 mmHg, and the residue was stored at 4 °C until further use.

### 2.3. Characterization of A.L. Extract

Preliminary phytochemical analysis was carried out, followed by the functional groups analysis present in the A.L. extract of *C. orchioides* were characterized using FTIR. The A.L. extract was subjected to FTIR (Bruker Optick Gm BH, Germany) characterization with a scan range from 400 to 4000 cm^−1^. The transmission percentage was recorded against the wavenumber. FTIR values were obtained in peaks, and functional groupings were predicted [25].

### 2.4. Fly Strain and Rearing

All the experiments in this study were performed using Oregon R+ (wild strain) flies and larvae. The flies were cultured at 24 ± 1 °C on a standard *Drosophila* cornmeal diet containing corn powder, sugar, yeast, agar, methylparaben, and propionic acid. The flies were maintained at a constant temperature of 24 ± 1 °C with 12-h light/dark cycles and relative humidity of 65–70% throughout the study.

### 2.5. Exposure of A.L. Extract to Drosophila

Four different concentrations (0.1, 1.0, 10.0, and 100.0 µg/mL) of *C. orchioides* A.L. extract were used for the study. The A.L. extract concentrations were selected based on a preliminary toxicity assessment. The A.L. extracts were mixed in the *Drosophila* diet to test toxicity in larvae and adults. The control group was retained in the same conditions for all experiments without treatment with A.L. extract.

### 2.6. Fly Developmental Assay and Phenotypic Analysis

First instar larvae were transferred to an untreated (control) and A.L. extract-treated medium (50 larvae/vial, 5 vials/group). The larvae were allowed to grow until they were ready to emerge as flies. The number of flies that emerged from untreated and treated groups was recorded until the emergence of the last fly in each group. The percentage of flies emerging from each group was evaluated (number of flies emerging from each group/the total number of larvae transferred to each group), and delay in emergence was also recorded. Additionally, post-exposure abnormalities in various body parts of the fly were examined using a Leica stereomicroscope (Model: S9D) [26].

### 2.7. Reproductive Performance

Reproductive performance of *Drosophila* was performed as described previously [27] with minor modifications. The newly emerged virgin male and female flies were separated and exposed to different concentrations of A.L. extracts for five days. After the fifth day, flies were paired and transferred to control food (10 pairs/group and one pair/vial). The number of eggs laid by each pair was counted for ten consecutive days, i.e., fecundity. Fertility was assessed by the number of offspring that emerged from each pair. Further, the percent hatchability was calculated as the fertility to fecundity ratio.

### 2.8. Climbing Assay

The locomotory behaviour of the flies was assessed according to instructions by Sharma et al. [26]. The newly emerged flies were subjected to A.L. extract for five days. After the fifth day, the flies were placed in empty cylindrical tubes with a height of 15 cm (20 flies/vial). The flies were then tapped down to the bottom and allowed 30 s to climb 15 cm from the bottom of the vial. The flies that crossed the 15 cm mark at 30 s were recorded. The experiment was performed in three trials/replicates. The climbing activity was represented as the performance index (P.I.).

### 2.9. Survival Assay

The effect of A.L. extract on the lifespan of adult flies was studied on newly emerged flies. The flies were transferred to untreated (control) and A.L. extract-treated medium from day one of their emergence (25 females and 25 males/vial and 5 vials/group). The flies were transferred to fresh untreated and A.L. extract-treated vials every day. Mortality was recorded in each vial until the last fly died on every alternative day [28].

### 2.10. RNA Isolation and Gene Expression Analysis

The total RNA was extracted from the control, and A.L. extract (100 µg/mL)-treated larval gut using the RNAiso Plus reagent (#9108, TAKARA, Japan). The purity and concentration of RNA were measured at an absorbance ratio of 260/280 and 230/260 nm using a MicroDrop spectrophotometer (Multiscan Sky, Thermo Scientific). Further, cDNA was synthesized using a cDNA synthesis kit (RR037A, TAKARA, Japan). qPCR (QuantStudioTM 5 Real-Time PCR system, Applied Biosystems) was performed in 96 well PCR plates using gene-specific primers (*hsps*) (primer details in Table 1) using power T.B. Green master mix (RR820A, TAKARA, Japan). The relative quantification of gene transcript expression was performed by amplifying *β-actin* as an internal control [29].

### 2.11. Quantitative Estimation of ROS

Reactive oxygen species (ROS) in the gut of the control and A.L. extract (100 treated larvae’s guts were measured using 2′, 7′-dichlorodihydrofluorescein diacetate, a fluorescent dye (DCFH-DA; Sigma, St. Louis, MO, USA, 2201608). The third instar larval guts from the control and A.L. extract-treated were incubated with 10 μM DCFH-DA for 45 min in the dark at room temperature. Further, the excess dye was removed by washing with 1X PBS three times (5 min. each), and the tissue was homogenized using 1X PBS. The absorbance was measured using a spectrofluorometer at 519 nm (Jasco, Tokyo, Japan, FP-8300). For representation, the stained larval guts were mounted and imaged using an Olympus BX53 fluorescent microscope [30]. The experiment was carried out with three biological replicates per group.

### 2.12. Immunostaining

The larval gut immunostaining was performed as described by D’Souza et al. [29]. The first instar larvae were transferred to control, and A.L. extract (100.0 µg/mL)-treated food. After 96 h, the guts of the larvae were dissected in 1X PBS, then fixed for 20 min at room temperature with a 4% paraformaldehyde (PFA) solution (made in 1XPBS). Following fixation, the guts were washed three times in 1XPBS for five minutes each. After washing, the gut tissues were blocked with 1% bovine serum albumin (BSA) for an hour. The larval guts were then incubated in primary antibody, anti-caspase 3 (1:500, #9661S, R&D systems), in 1% BSA at 4 °C overnight. The guts were then washed with 1XPBST (0.3 percent Triton X-100 in 1XPBS) and incubated in secondary antibody (#A11030; Goat anti-mouse Alexa flourTM 546; 1:500 dilution) for 3 h. The excess secondary antibody was washed three times with 1XPBST, followed by DAPI staining for 30 min. The guts were then mounted on a slide and analyzed using Olympus BX53 fluorescent microscope. At least three independent biological replicates per group were used to perform the experiment.

### 2.13. Statistical Analysis

Statistical analysis was performed using the Prism software (GraphPad version 8.4, San Diego, CA, USA). The statistical significance of the mean values for different parameters was monitored in control and treated flies using one-way ANOVA. By using the Gehan-Breslow-Wilcoxon test, the significance of the survivability assay was determined. *p*-values represented as *p* < 0.05 (*), *p* < 0.01 (**), *p* < 0.001 (***) and *p* < 0.0001 (****).

## 3. Results and Discussion

This study aimed to determine the toxicity of A.L. extract at the cellular and organismal levels using *D. melanogaster.* Our preliminary results of the acute exposure (24 h) indicated (data not shown) that the oral exposure of 500, 1000, and 2000 µg/mL of *C. orchioides* A.L. extract to *Drosophila* larvae did not cause any organismal death. Hence 100.0 (1/20 of 2000 µg/mL), 10.0 (1/200 of 2000 µg/mL), 1.0 (1/2000 of 2000 µg/mL), 0.1 (1/20,000 of 2000 µg/mL) µg/mL concentrations were selected for the study.

### 3.1. Characterization of A.L. Extract Using FTIR

Plants contain various components that have often been used to make natural medicines. Because of the unparalleled richness of chemical compounds, plant-derived natural products, as a standardized extract or in pure form, give limitless prospects for new therapeutic leads [31]. The present study used FTIR to identify the functional groups in A.L. extracts of *C. orchioides* post preliminary phytochemical analysis. The peak value and the functional group are displayed in Table 2. The characteristic absorption bands were exhibited at 618.51 cm^−1^, 648.01 cm^−1^, 696.43 cm^−1^, 721.35 cm^−1^, 1017.59 cm^−1^, 1453.90 cm^−1^, 1652.33 cm^−1^, 3365.12 cm^−1^, 3609.93 cm^−1^, 3713.81 cm^−1^, 3887.79 cm^−1^, and 3941.77 cm^−1^. The presence of the N-H stretch at 3365.12 cm^−1^ indicates the presence of alkaloids in the extract [32]. The characteristic stretching band of O-H was observed at 3609.93 cm^−1^, 3713.81 cm^−1^, 3887.79 cm^−1^, and 3941.77 cm^−1^, indicating the presence of phenolic compounds (Figure 1). In agreement with Umar et al. [32], the ethanolic leaf extracts of *C. orchioides* showed a broad absorption spectrum of O-H stretch at the region of 3300 cm^−1^, indicating the presence of the phenolic group. Additionally, C-O bending vibrations were observed in the fingerprint region, indicating the presence of an alcohol functional group similar to our findings. These studies indicate that the A.L. extract is rich in chemical constituents of alkaloids, phenolic, and saponins; implicating their attributes to the reported bioactivity.

### 3.2. A.L. Extract Supplementation Causes Developmental Toxicity and Wing Deformity in Drosophila

The organism’s developmental period is critical and susceptible to physical/biological/chemical stress as stress could hamper an organism’s development program, leading to lifelong damage to the adults. Supplementation of A.L. extracts to the early larval stage negatively impacted the development of *Drosophila* larvae (Figure 2). As shown in Figure 2a–h, no deleterious effect on the organism’s development was recorded at the lowest concentration of 0.1 µg/mL. With increasing concentrations of A.L. (1.0, 10.0, and 100.0 µg/mL), the organism demonstrated a progressive and significant reduction in emergence of 17.6 % (*p* < 0.0001), 25.6% (*p* < 0.0001) and 29.2% (*p* < 0.0001), respectively, specifying the developmental toxicity (Figure 2a). However, no delay in emergence was observed in any tested A.L. concentrations (Figure 2b). Furthermore, 1.0, 10.0, and 100.0 µg/mL concentrations of A.L. extracts showed aberrant wing malformations like wrinkled wings and crushed wings (Figure 2c–g) at the frequency of 21.2, 39.6, and 46.4%, at 1.0, 10.0 and 100 µg/mL (Figure 2h), respectively, thus inferring the teratogenic potentiality of A.L. extract. Mounting evidence involving in vitro and in vivo models has shown that plant alkaloids can cause developmental defects as reviewed by Green et al., [33]. Alkaloids impact various animal metabolic systems, and their harmful mechanisms of action might differ significantly. Toxicity can be caused by enzymatic changes that disrupt physiological processes, intercalating with nucleic acids to limit DNA synthesis and repair mechanisms, or influencing the nervous system. Several alkaloids have the potential to alter a host of biological functions [34] such as the study wherein [35] supplementation of the A.L. extracts of *Vinca rosea* to *Drosophila* resulted in significant developmental defects like reduction and delay in emergence and wing deformity. This finding aligns with the previous study on the A.L. extracts of *Ruta graveolens* at concentrations of 5, 10, and 20% when orally administered to rats for four days, hampered the preimplantation development and embryo transport and also resulted in abnormal embryos [36]. Similarly, A.L. extract of *Ficus glomerata* at a concentration of 125, 250, and 500 ppm, when administered to *zebrafish* causes morphological, and developmental abnormalities such as delayed growth, decreased heart rate, and decreased body length [37]. In corroboration with the studies mentioned above, our results demonstrate A.L. extract-induced developmental toxicity in the organism.

### 3.3. A.L. Extract Moderates the Drosophila Reproductive Performance

The reproductive value of flies was measured using two fitness parameters: fecundity and fertility, which can be influenced by factors like fly genotype, body size, age, and mate, as well as environmental conditions [38]. In the present study, supplementation of A.L. extract to adult flies significantly affected the reproductive performance of the organism in a dose-dependent manner (Figure 3). For instance, statistically significant reductions in fecundity of 22.92% (*p* < 0.0001), 28.84% (*p* < 0.0001), and 33.53% (*p* < 0.0001) at 1.0, 10.0, and 100 µg/mL were observed (Figure 3a) and decline in fertility by 66% (*p* < 0.0001) and 57% (*p* < 0.0001) decline in fecundity and fertility were observed upon 100.0 µg/mL A.L. extract exposure (Figure 3a,b). The percentage hatchability (fertility/fecundity) was 95, 86, 82, and 79% at 0.1, 1.0, 10.0, and 100.0 µg/mL A.L. extract, respectively, compared to the control (Figure 3c). The presence of several potential active substances such as phenolics, flavonoids, alkaloids and saponins, and polysaccharides were demonstrated in different extracts of *C. orchioides* [13,39,40], and from our FTIR results, it is clear that phenolics and alkaloids are present. Orally administered *Chromolaena odorata* leaf alkaloids inhibit gonadotropin release in male rats resulting in testosterone deprivation and aberrant spermatozoa. Further, the secretory and synthetic functions of the testes and sperm were affected [41]. Similarly, oral administration of *Cortex albiziae* saponins altered the structure of the ovary and uterus, decreasing pregnancy rates [42]. Recently we reported that the A.L. extract of *C. orchioides* contains alkaloids and saponins at 2.64 g/100 g and 3.00 g/100 g, respectively, which might be a possible reason for the poor reproductive performance of the organism upon A.L. exposure [20]. Moreover, the results of our study point toward reproductive toxicity rendered by the synergistic effect of alkaloids and saponins present in the A.L. extract. Our results are corroborated by a previous study on mice [43]. It is also important to underline that the Ayurvedic and Chinese traditional system of medicine claims that the rhizome extracts of *C. orchioides* can be used as a tonic for strength, vigour, and vitality; finding application in several restorative and aphrodisiac formulations [14]. Few reports have evidenced that the rhizome extracts of *C. orchioides* can also enhance sexual activity in male rats by increasing spermatogenesis. The rhizome extract of *C. orchioides* also has estrogenic activity [44]. However, rhizome extracts or whole plants have been used in previous studies. The present study explores the leaf extract explicitly, thus, implicating the aqueous leaf extracts for reproductive toxicity in *Drosophila*.

### 3.4. A.L. Extract Supplementation Does Not Affect the Locomotory Behaviour of the Exposed Organism

*Drosophila* has a well-developed nervous system and offers several benefits to studying the nerve physiology of the behavioural traits and the endpoints of genetic and environmental factors [45]. Locomotion is a very robust motor pattern that represents the health of an organism’s neuronal system. In the current study, the climbing ability of the organism after A.L. extract exposure for five days was assessed, and the results showed an insignificant change in the climbing behaviour of the organism compared to the control (Figure 4). Inadequate availability of A.L. extract to the brain due to the blood-brain barrier or short exposure period could be a reason for the non-apparent change in the organism’s climbing behaviour. An earlier report has shown that Curculigoside, an active component of *C. orchioides*, protects the neurons from N-methyl-D-aspartate (NMDA) induced toxicity by decreasing the apoptotic proteins and by reducing the production of intracellular ROS in cultured cortical neurons [15]. Besides, Curculigoside isolated from rhizomes of the plant exhibits antidepressant activity in mice. It causes a significant increase in dopamine, norepinephrine, and 5-hydroxytryptamine levels, leading to the up-regulation of brain-derived neurotrophic factor proteins in the hippocampus of chronic mild-stress rats [46].

### 3.5. A.L. Extract Decreases Lifespan in Flies

Studies have shown that many herbal extracts are known to extend lifespan [47,48]. Hence, we studied the effect of A.L. extract on the fly’s lifespan. After adult eclosion, the flies were fed with A.L. extract of 0.1, 1.0, 10.0, and 100.0 µg/mL. The decline in the survival of adult flies was found to be dose-dependent in A.L. treated groups (Figure 5). The survival of organisms at the lower concentrations (0.1 and 1.0 µg/mL) (mean and percentage survival) was comparable to the control, whereas reduced survival was observed at higher concentrations. The control, 10.0 and 100.0 µg/mL A.L. extract-treated flies showed a mean survival of 53, 47, and 41 days, respectively. The percent reduction in survival at 10.0 and 100.0 µg/mL A.L. extract was found to be 10.32% (*p* < 0.01) and 22.65% (*p* < 0.0001), respectively (Figure 5a,b). In previous studies, the exposure of harmala alkaloids showed a reduced life span in *Tribolium castaneum* and *Rhizopertha dominica* [49]. The alkaloids present in the aqueous extract of *C. orchioides* probably affected the organism’s lifespan.

### 3.6. A.L. Extract Does Not Elevate Cellular Stress in Drosophila Larval Gut

Cells have evolved multiple stress response pathways to protect cellular homeostasis under varying environmental and physiological situations. Among various stress response pathways, the upregulation of Hsps has been considered a first-tier indicator against different stress conditions [50]. Since the gut is the primary site for xenobiotic exposure, the effect of the extract at the cellular level was studied using the gut tissue. To examine whether exposure to A.L. extract causes cellular stress, we evaluated the expression of small *hsps* (22, 26, and 27) and large *hsps* (70 and 83) in the larval gut after 96 h of A.L. extract (100.0 µg/mL) exposure. However, none of the *hsps* expressions was modulated significantly after the treatment of A.L. extract (Figure 6a). Apart from *hsps*, oxidative stress is one of the critical mediators of cellular toxicity upon xenobiotic exposure [51]. Therefore, the ROS level was estimated in the gut of the A.L. (100.0 µg/mL) exposed (96 h) larvae. Similar to the *hsp* expression, the ROS levels in the A.L. exposed gut were comparable to the ROS levels of the control gut (Figure 6b,c). Activation of cell death is the primary mode of mechanism during toxicity. Mounting evidence from scientific studies suggests that plant extracts activate cell death in various health and disease conditions [52,53]. Hence, we evaluated the effect of A.L. extract in inducing cell death. Antibody staining against cleaved caspase-3 in larval gut indicated no statistically significant changes in cleaved caspase-3 expression upon A.L. extract exposure compared to the control (Figure 6d). Our findings suggest that insignificant changes in the cellular stress markers and considerable no cell death suggest that A.L. extract might not be toxic to the cells after short-term exposure.

## 4. Conclusions

This study concludes that exposure to A.L. extract of *C. orchioides* causes organismal level toxicity in *Drosophila*, evidenced by developmental defects, poor reproduction performance, and reduced organism survival. Furthermore, alkaloids, phenols, and saponins were found in the A.L. extract of *C. orchioides*, which could be a plausible reason for the observed toxicity. However, the toxicity assessment of the individual compound needs further investigation. The study concludes with an expression of concern for the long-term use of *C. orchioides* extract for therapeutic purposes.

## Figures and Tables

**Figure 1 ijerph-19-15218-f001:**
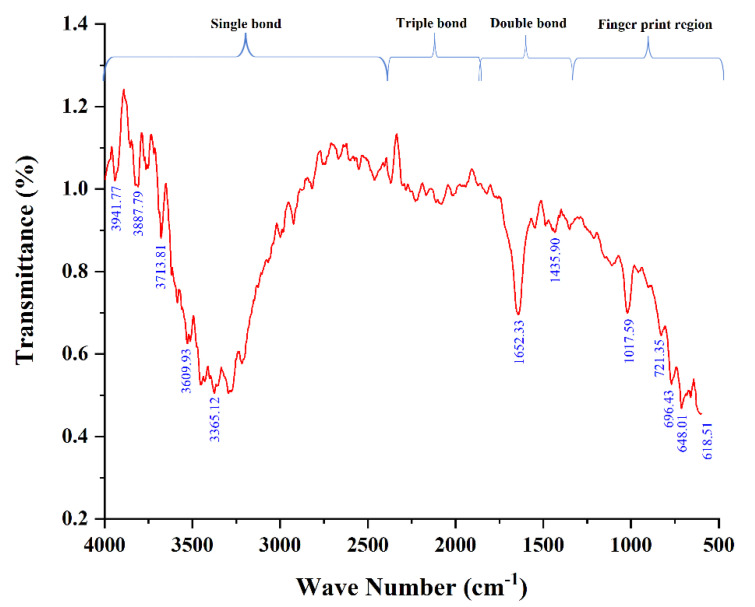
FTIR spectra of the *C. orchioides* A.L. extract.

**Figure 2 ijerph-19-15218-f002:**
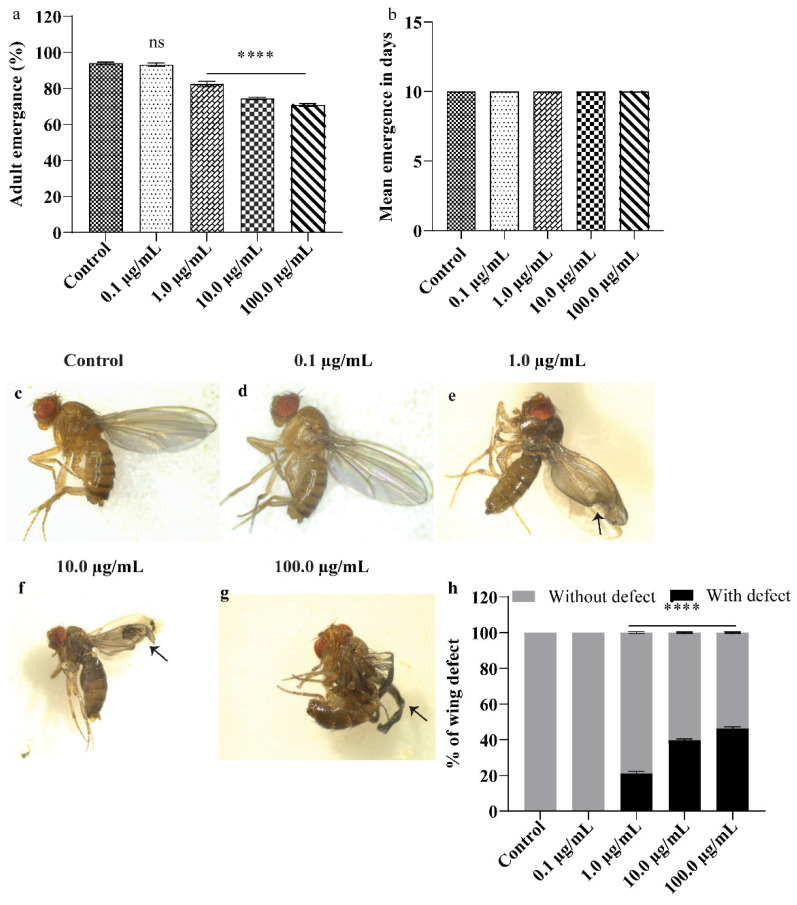
Developmental assay. (**a**) Total percentage of adult emergence in control and A.L. extract-treated groups (**b**) Mean emergence in days (**c**–**g**) representative image of flies with wing phenotypes with control and A.L. extract-treated flies (**h**) percentage of wing defect. Significance is ascribed as ns and **** *p* < 0.0001 compared to control.

**Figure 3 ijerph-19-15218-f003:**
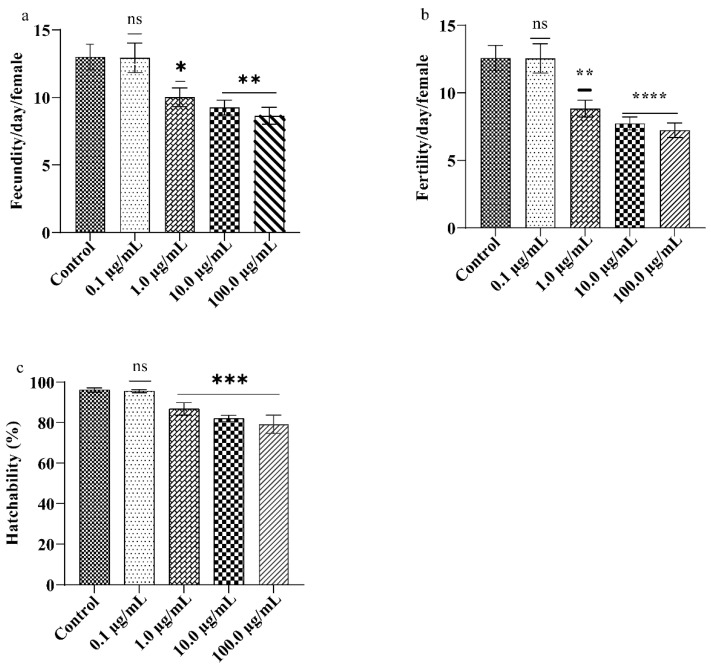
Reproductive assay: (**a**) fecundity, (**b**) fertility, (**c**) percentage hatchability. Significance is ascribed as ns (non-significant), * *p* < 0.05, ** *p* < 0.01, *** *p* < 0.001 and **** *p* < 0.0001 with respect to control.

**Figure 4 ijerph-19-15218-f004:**
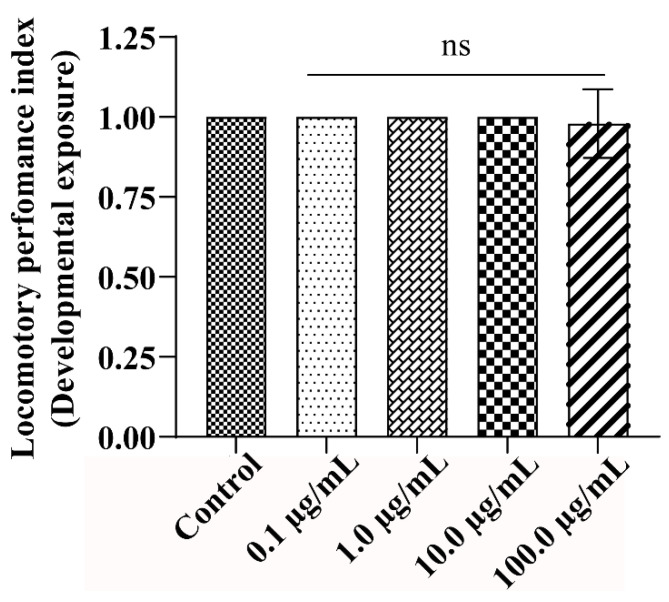
Behavioural assay: Jumping capability of flies eclosed from developmental exposure of larvae fed on control and A.L. extract-treated food. The graph indicates no climbing defect upon exposure to A.L. extract treatment. Significance is as ascribed as ns (non-significant).

**Figure 5 ijerph-19-15218-f005:**
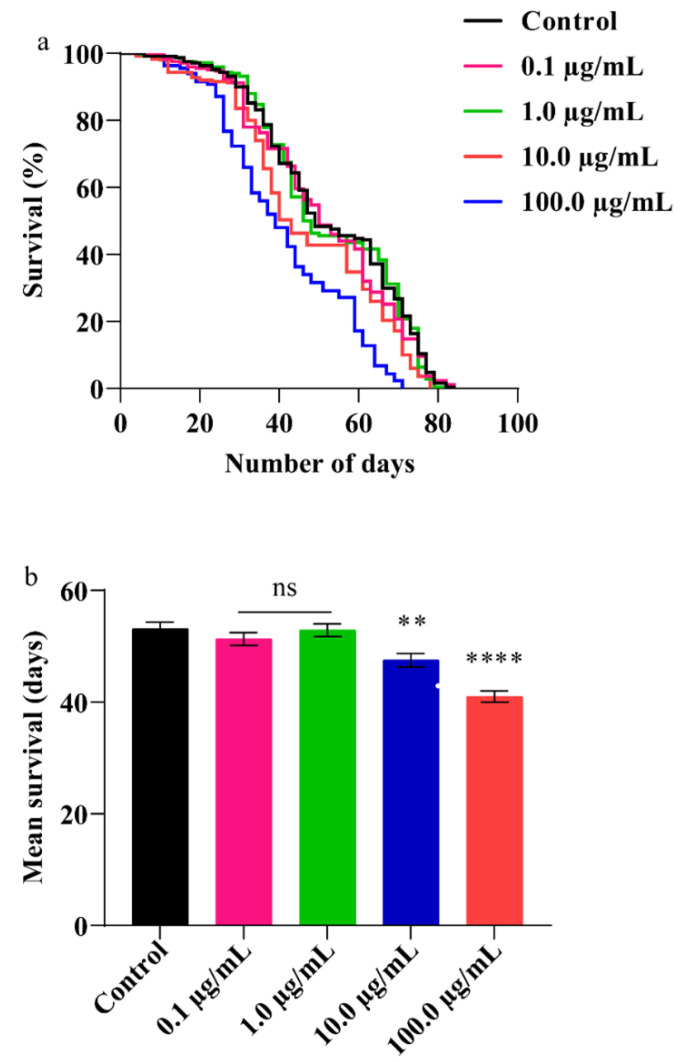
Survival assay: (**a**) Survival of flies in percentage (**b**) Mean survival of flies. Significance is ascribed as ns (non-significant), ** *p* < 0.01, **** *p* < 0.0001 with respect to control.

**Figure 6 ijerph-19-15218-f006:**
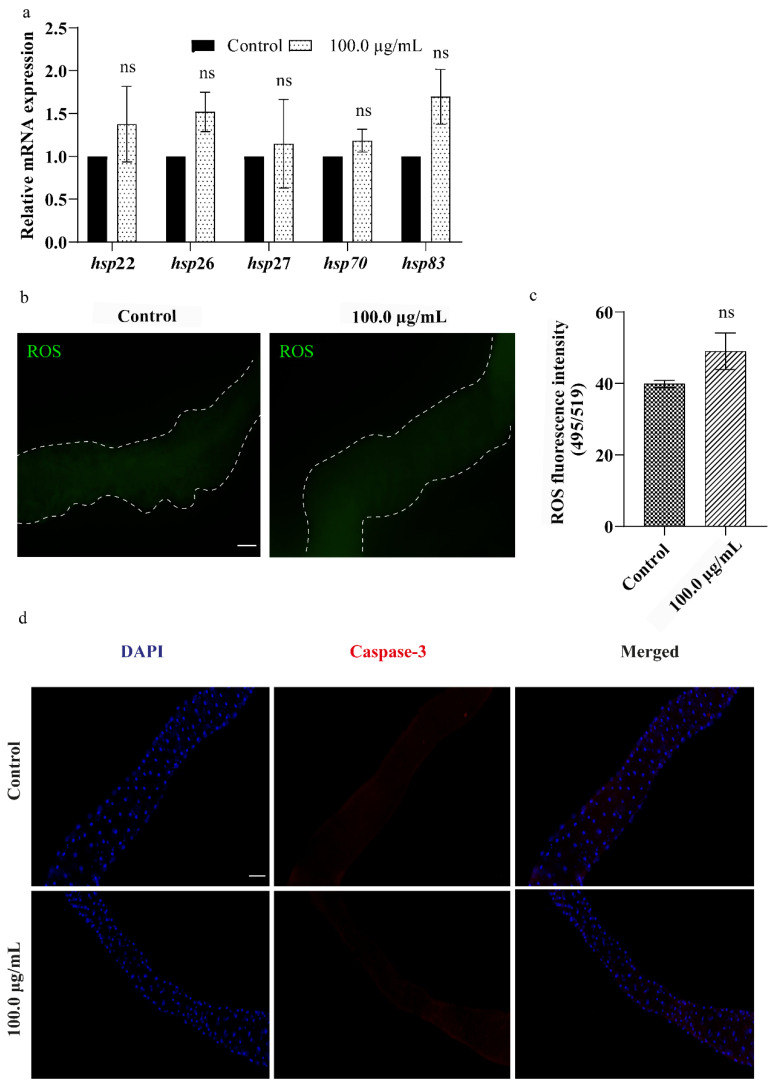
Effect of A.L. extract of *C.orchioides* at cellular level: (**a**) *hsps* expression. The graph represents non-significant *hsps* expression in the gut of larvae exposed to A.L. extract. (**b**,**c**) ROS generation. The graph represents no major change in intensity of DCF fluorescence in control and A.L. extract-treated groups. (**d**) Apoptotic assay. The graph illustrates no cell death observed in the gut of control larvae and the highest concentration of A.L. extract-fed larval gut (100.0 µg/mL). The gut tissues are stained with DAPI. Significance is ascribed as ns *p* > 0.05. The scale bar corresponds to 100 µm.

**Table 1 ijerph-19-15218-t001:** Primer details.

Gene	Sequence (5′-3′)
*hsp22*	Forward primer: TGGCTATAGCTCCAGGCACT Reverse primer: GCTTTGTCATTTGGCTCCTC
*hsp26*	Forward primer: GAGCGCATCATTCAAATTCA Reverse primer: TCCACACCAGGTGAACAAAA
*hsp27*	Forward primer: GACTGGGTCGTCGTCGTTAT Reverse primer: TTGAACTGCGACACATCCAT
*hsp70*	Forward primer: CATTCCGTGCAAGCAGACTA Reverse primer: GCTGACGTTCAGGATTCCAT
*hsp83*	Forward primer: CAGCTGGTCTCTGTCACCAA Reverse primer: TGGACTTCATCAGCTTGCAC
*β-Actin*	Forward primer: GTGCCCATCTACGAGGGTTA Reverse primer: AGGGCAACATAGCAGCTT

**Table 2 ijerph-19-15218-t002:** FTIR peak values with functional group of *C. orchioides* A.L. extract.

Frequency Range (cm^−1^)	Frequency Peak (cm^−1^)	Bond	Functional Group
600–650	618.51	C-Br stretch	Alkyl halides
	648.01		
680–720	696.43	C-H “oop”	Aromatics
	721.35	C-H rock	Alkanes
1000–1500	1017.59	C-O stretch	Alcohols, carboxylic acids
	1453.90	C-C stretch	Aromatics
1600–1700	1652.33	C=C	Alkenes
3300–3610	3365.12	N-H stretch	1° Amines
	3609.93		
3700–4000	3713.81	O-H stretch, free hydroxyl	Alcohols, Phenols
	3887.79		
	3941.77		

## Data Availability

The data that support the findings are available with corresponding authors upon reasonable request.

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
