# Peer review of "Toxicity Assessment of Curculigo orchioides Leaf Extract Using Drosophila melanogaster: A Preliminary Study"

_ijerph, 2022, doi:10.3390/ijerph192215218_

Round 1

Reviewer 1 Report

This is an interesting study showing the toxicity of aqueous leaf extract of Curculigo orchioides in Drosophila melanogaster.  Although authors have tried to provide a lot of data, few things need to be addressed.

Comment 1: Please use italics text for botanical and zoological names. In the title and text Curculigo orchioides and Drosophila melanogaster are not italicized.

Comment 2: For ROS measurement it would be better to use fluorescent staining instead of measuring in homogenate.

Comment 3: Did authors examine the gut permeability in control and A.L. extract treated larvae to check if A.L. extract of Curculigo orchioides causes any change in gut permeability?

Comment 4: It would be interesting to examine which phytochemical present in A.L. extract of Curculigo orchioides is responsible for toxicity in Drosophila melanogaster. Purification of phytochemicals and toxicity assessment will provide a clear understanding.

Comment 5: Current study shows that A.L. extract of Curculigo orchioides caused an array of abnormalities in Drosophila melanogaster. What would be the underlying mechanism of toxicity caused by A.L. extract of Curculigo orchioides in Drosophila melanogaster as there is no cell death and oxidative stress after treatment of A.L. extract of Curculigo orchioides?

Author Response

Respected Reviewer 1

We thank you for the constructive and critical suggestions for improving our manuscript. We have revised the same in light of the reviewers’ comments. All changes made in the revised manuscript are made in track changes.

Reviewer 2 Report

Reviewer report for manuscript  IJERPH-1993209

The manuscript titled "Toxicity assessment of Curculigo orchioides leaf extract using Drosophila melanogaster" aimed to evaluate the aqueous extract from Curculigo orchioides regarding its toxic effect in Drosophila melanogaster (including teratogenic potential, reproductive performance, locomotor behavior, lifespan, and others). The authors reported interesting results, suggesting that the popular use of this extract for the therapeutic purpose may lead to relevant adverse effects. The manuscript is well-written and requires minor corrections. The major comment is regarding the lack of experimental data to support the conclusion of the extract composition. This resulted in potential misconclusions, where the toxic effect observed in different animal models exposed to other extracts (some of them prepared by different methodology from the one reported here) were extrapolated to Drosophila melanogaster without any experimental data to support that the same compound indeed caused the observed effect. Given the preliminary nature of this study, the reviewer suggests the addition of a subtitle indicating this status (e.g., "Toxicity assessment of Curculigo orchioides leaf extract using  Drosophila melanogaster: A preliminary study") or changing the manuscript to a short communication. After performing the suggestions for improvement, the manuscript should be suitable for publication.

Major Comments

The use of  FTIR as the only characterization method to evaluate the A.L. extract composition seems to provide minor contributions in this regard. As observed in section 3.1, this method is limited to identifying functional groups (e.g., alkenes, alkanes, alcohols, carboxylic acids), which are common groups in a broad range of organic compounds. As a result, it is not possible, based on the given data, to conclude that "the A.L. extract is rich in chemical constituents of alkaloids, phenolic and saponins." (lines 211-212) since these groups might be from other organic compounds displaying the same functional groups. It is not possible to distinguish them only with one FTIR spectrum. Additionally, not all compounds in the same group (e.g., alkaloid) are responsible for a given toxic effect (e.g. wing malformation). Therefore, conclusions like "our results suggest that the presence of alkaloids and saponins may be responsible for A.L. extract-induced developmental toxicity in the organism" (lines 246 – 248) should be replaced by the actual substance responsible for this effect. The reviewer understands that this is not an easy request. The main point here is to highlight that the manuscript is incomplete. As a result, it is suggested to add a subtitle indicating this status (e.g., "Toxicity assessment of Curculigo orchioides leaf extract using  Drosophila melanogaster: A preliminary study") or publishing it as a short communication;

      Furthermore, conclusions such as "our observation supports previous studies, wherein the exposure of harmala alkaloids showed a reduced life span in Tribolium castaneum  and Rhizopertha dominica" (lines  320- 321) should not be made; since there are neither experimental data to support the claim that the alkaloid obtained from an aqueous extract of C. orchioides is the same from a methanolic extract of Peganum harmala. Nor that the observed toxic effect in Drosophila melanogaster can be extrapolated to Tribolium castaneum and Rhizopertha dominica. The same idea applies to ethanolic extracts of Trapa natans leaf in Zebrafish embryos (239 – 240), Clematis aethusifolia in larvae of  Plutella xylostella (243-244), Chromolaena odorata leaf alkaloids in male rats (268 – 269), among others.

Minor comments

1.      Line 15 – Please, replace "using the Drosophila" with "using Drosophila melanogaster as an experimental model" 

2.      There are reports in the literature that Curculigo orchioides is currently listed as an endangered species (https://doi.org/10.1007/s11627-021-10246-5). This information should be added; 

3.      The misconception of plant-based medicine being non-toxic is mainly a cultural understanding. Therefore, it seems to be more appropriate to replace the term "commoner" with a more inclusive one (line 43);

4.      To avoid unnecessary repetition, please review the following statement: "the supplementation of A.L. extract to  Drosophila  larvae for  96hrs does not elicit gut toxicity, however, the supplementation of A.L. extract to larvae or adult flies significantly hinders the development, reproduction, and survival of the organism." (lines 78- 80); 

5.      2. Materials and Methods – The authors are encouraged to add a section named "materials" and report all the materials used to perform the experiments and analysis, along with their respective suppliers. This is a relevant topic to ensure appropriate reproducibility; 

6.      2.1. Collection of plant material and extraction –  As it is well-known, plant secondary metabolites are strongly dependent on a variety of factors such as the environment (e.g., light, temperature, soil water) and the physiological condition of the plant (e.g., flowering, presence of microorganism infection)  (https://doi.org/10.3390/molecules23040762, https://doi.org/10.3389/fpls.2021.621276). Therefore, please better describe the plant's environmental and physiological condition when they were collected. Moreover, to ensure reproducibility, report (I) the proportion water:mass of the leaves, (II) the specifications of the instrument used during the shaking process, (III)  the operation conditions of the flash evaporator (e.g., temperature, pressure). Furthermore, The paper of Aloysius et al. (2020) is not included in the reference list; 

7.      2.2. Characterization of A.L. extract – To carry out the FTIR analysis, were the samples mixed with KBr? If affirmative, please inform and report the proportion KBr: sample, the relative humidity (the water content may affect this analysis), the resolution, and the number of scans carried out for each sample; 

8.      2.3. Fly strain and rearing – Please, report how the flies were obtained; 

9.      2.4. Exposure of A.L. extract to Drosophila – It is unclear if this section is a previous description of the conditions used in the following experiments (e.g., 2.5-2.8) or an independent experiment. If the former one, please inform it. It the last one, better describe it (e.g., for instance, (I) did the housing conditions change from the ones described in section 2.3? (II) the number of individuals in each group, (III) the time of exposure, (IV) the larval stage) and inform if the individuals (larvae or flies) in sections 2.5 - 2.8 were exposed to the same concentrations of C. orchioides A.L. extract reported in this section 2.4. If  not, report the concentration used; 

10.  2.5. Fly developmental assay and phenotypic analysis – Inform the criteria used to classify the time to emerge as delayed and, from this point, how it was calculated. Moreover, did the flies were transferred daily to vials containing fresh untreated and A.L. extract-treated vials as reported in section 2.8? (line 139) ; 

11.  2.6. Reproductive performance – Inform how the flies were captured to perform the pairing; 

12.  3. Results and Discussion – To avoid unnecessary repetition, please remove the first paragraph (lines 183 – 194) for the topics mentioned were already highlighted in the introduction; 

13.  3.5. A.L. extract decreases lifespan in flies – It was reported that the flies (without treatment and exposed to 10.0 and 100.0 µg/mL A.L. extract) exhibited mean survival of 53, 47, and 41 days, respectively (lines 317-318). Afterward, it was stated that the percentage reduction in survival was 47.5 and 41.0 in the groups treated with 10.0 and 100.0 µ g/mL, respectively. It is not clear how these values were calculated since the difference between the treated groups, and the control is  6 and 12 days, respectively to 10.0 and 100.0µ g/mL, which represents a relative percentage reduction of ~11% (i.e., 6/53*100) and ~23% (i.e., 12/53*100) respectively. Please, better explain how these values were calculated; 

14.  It seems that in the statement "we observed the  effect of A.L. extract in inducing cell death." (line 343) The term "evaluated" suits better than "observed" as no effect of the extract on inducing cell death was observed (lines 343-346); 

15.  Please, add references to support the following statement:

a)      Lines 40: 42 – "Factors such as low-cost alternatives to allopathic medicine, efficacy, easy availability, and the belief in  fewer side effects are the most common reason for using herbs as medicine." 

b)      Lines 71-72 – "Drosophila is the closest invertebrate model to humans, with conserved evolutionary genetics  and  developmental  biology" 

c)      Line 277 – "Our results are in corroboration by a previous study on mice" 

d)      Section 3.4 (lines 299 – 305) – Curculigoside is a compound in the A.L. extract, and its protective neuronal effect observed in mice and rats can be extrapolated to Drosophila melanogaster.

Author Response

Respected Reviewer

We thank you for the constructive and critical suggestions towards improving our manuscript. We have revised the manuscript as per the comments. All changes made in the revised manuscript are made in track changes for your kind perusal.

Reviewer 3 Report

The reviewer's recommendations to the authors are listed in the comments in the attached manuscript.

Author Response

Dear Reviewer 3

We thank you for the suggestions towards improving the manuscript. We have modified the manuscript and accepted all proposed changes per the suggestion. All changes made in the revised manuscript are made in track changes for your kind perusal. Kindly accept and oblige.